# Multipath/NLOS Detection Based on K-Means Clustering for GNSS/INS Tightly Coupled System in Urban Areas

**DOI:** 10.3390/mi13071128

**Published:** 2022-07-17

**Authors:** Hao Wang, Shuguo Pan, Wang Gao, Yan Xia, Chun Ma

**Affiliations:** 1School of Instrument Science and Engineering, Southeast University, Nanjing 210096, China; whao@seu.edu.cn (H.W.); gaow@seu.edu.cn (W.G.); machun@seu.edu.cn (C.M.); 2Key Laboratory of Micro-Inertial Instrument and Advanced Navigation Technology, Southeast University, Nanjing 210096, China; 3School of Geomatics Science and Technology, Nanjing Tech University, Nanjing 211800, China; yanx@njtech.edu.cn

**Keywords:** GNSS/INS tightly coupled system, multipath/NLOS detection, K-means clustering algorithm, urban areas

## Abstract

Due to the massive multipath effects and non-line-of-sight (NLOS) signal receptions, the accuracy and reliability of GNSS positioning solution can be severely degraded in a highly urbanized area, which has a negative impact on the performance of GNSS/INS integrated navigation. Therefore, this paper proposes a multipath/NLOS detection method based on the K-means clustering algorithm for vehicle GNSS/INS integrated positioning. It comprehensively considers different feature parameters derived from GNSS raw observations, such as the satellite-elevation angle, carrier-to-noise ratio, pseudorange residual, and pseudorange rate consistency to effectively classify GNSS signals. In view of the influence of different GNSS signals on positioning results, the K-means clustering algorithm is exploited to divide the observation data into two main categories: direct signals and indirect signals (including multipath and NLOS signals). Then, the multipath/NLOS signal is separated from the observation data. Finally, this paper uses the measured vehicle GNSS/INS observation data, including offline dataset and online dataset, to verify the accuracy of signal classification based on double-differenced pseudorange positioning. A series of experiments conducted in typical urban scenarios demonstrate that the proposed method could ameliorate the positioning accuracy significantly compared with the conventional GNSS/INS integrated navigation. After excluding GNSS outliers, the positioning accuracy of the offline dataset is improved by 16% and 85% in the horizontal and vertical directions, respectively, and the positioning accuracy of the online dataset is improved by 21% and 41% in the two directions. This method does not rely on external geographic information data and other sensors, which has better practicability and environmental adaptability.

## 1. Introduction

Global Navigation Satellite System (GNSS) can provide all-day and all-weather global Positioning, Navigation, and Timing (PNT) services for global users, and its positioning errors will not accumulate over time [1,2]. However, satellite signals are frequently blocked or even lose lock in complex urban scenarios, which cannot guarantee the effectiveness of positioning [3]. Inertial Navigation System (INS) has the advantages of strong autonomy and strong anti-interference and can obtain short-term, high-precision navigation and positioning results [4]. However, INS errors accumulate over time, and long-term independent solution can result in reduced accuracy or even divergence [5]. GNSS and INS possess highly complementary characteristics, which can effectively overcome the adverse effects of the harsh environment with the two systems integrated [6]. Therefore, GNSS/INS integrated navigation is widely utilized in naturalistic driving, high-precision vehicle navigation, Intelligent Transportation system (ITS), and autonomous driving [7,8,9,10]. Based on the integrated method, GNSS/INS integrated navigation can be divided into loosely coupled, tightly coupled, and deeply coupled. Among them, the GNSS/INS tightly coupled system can maintain continuous positioning despite the insufficiently visible satellites and has the advantage of simple integrated structure and easier implementation, which has been extensively recognized by many scholars [11,12,13].

In the application of vehicle dynamic positioning, a continuous, reliable, and high-precision positioning method is urgently required. Balsa-Barreiro (2013) implemented an innovative methodology based on vehicle speed for the geo-referencing naturalistic driving method, which can overcome the problems related to the lack of positioning data [14]. However, this method requires extra geographic information. In addition, some scholars use GNSS and multiple sensors for positioning, which can integrate the advantages of each sensor, and obtain continuous and reliable positioning information in urban scenes [15,16,17]. But, the cost of multi-sensor is high, the weight is difficult to determine, and the amount of calculation is large. Therefore, based on the above discussion, this paper chooses GNSS/INS tightly coupled system positioning as the method to obtain the vehicle position in urban scenarios. However, the performance of GNSS/INS tightly coupled positioning is inevitably plagued by various outliers in the complex urban scenario [18]. For the GNSS receiver, the multipath effect and NLOS signal are the commonly adverse factors to restrict the positioning performance of GNSS, which further subsides the GNSS/INS integration system positioning capability [19,20]. Multipath effect occurs when a signal is received through multiple paths between the satellite and the receiver antenna. Additionally, multipath interference can affect the phase detection characteristics of the receiver tracking loop, resulting in tracking and measurement errors [21]. Multipath includes both direct and reflected signals, and the reflected signals can be multiple. NLOS signal reception occurs when the direct path from the satellite to the receiver is blocked and the signal can only be received through the reflected path [22]. Due to the relatively low cost and the ability to provide accurate time reference and absolute coordinate information, GNSS solution is still an important and preferred technical method in the field of high-precision navigation and location services. Unfortunately, the degradation of GNSS positioning accuracy may negatively affect the performance of the entire system [23]. We need to mitigate the error of the standalone GNSS positioning with innovative signal-processing methods to promote the performance of GNSS/INS integration. 

Accurate multipath/NLOS detection and subsequent processing are the basis for the signal quality control and positioning strategy optimization of the GNSS/INS integration Navigation system [24]. With the prosperity of artificial intelligence technology, more and more scholars began to employ machine learning or deep-learning methods to identify multipath/NLOS signals. These methods constructed the mapping relationship between multiple feature parameters and GNSS signal categories to reduce the operating cost of traditional methods and enhance the availability of the algorithm, which have achieved outstanding results [25,26,27,28]. The above approaches are classifiers based on supervised learning that need to label the training samples in advance. Meanwhile, the accuracy of the labeling is directly associated with the performance of the classifiers. In practical applications, it is challenging and expensive to obtain an accurately labeled dataset that covers multiple scenarios and represents all state types. In order to refrain from the limitations of signal classification methods caused by the above problems, scholars have begun to use unsupervised learning techniques for signal classification research, which can sufficiently excavate the information of the observation data itself [29,30,31,32]. Compared with the supervised-learning multipath/NLOS signal detection algorithm, the unsupervised technology has better advantages in availability and environmental applicability. Furthermore, the positioning accuracy after excluding contaminated GNSS satellites is significantly better than the traditional threshold method and the classic RAIM algorithm. 

Given that the multipath/NLOS signal restricts the dynamic positioning accuracy and reliability of the GNSS/INS integrated system, apart from GNSS alone positioning, multipath/NLOS detection and mitigation for GNSS/INS integrated systems were developed in past years [33,34,35]. However, these technologies are dependent on geographic information data or external sensor equipment, such as 3D building models, cameras, and LiDAR, which have a certain level of deficiencies in terms of availability, cost, and security. In addition, the equivalent weight model is employed to construct robust algorithms that can weaken the influence of the gross error of the observation on the positioning accuracy [36,37]. This method utilizes the robust factor to adjust the filter gain moment or the observation noise for GNSS/INS integrated positioning, which plays a role in suppressing multipath and NLOS errors to a certain extent. However, the robust estimation algorithm has difficulty handling multiple outliers on the same epoch and relies heavily on the correctness of the robust model. Therefore, when the original observations are seriously polluted by the multipath/NLOS signals in harsh environments, the reliability of the algorithm cannot be guaranteed.

This paper aims to further promote the accuracy of GNSS/INS tightly coupled positioning results by using unsupervised techniques to detect multipath/NLOS signals. A clustering algorithm is utilized to label GNSS data in offline system as normal and abnormal observations, the latter mainly caused by multipath/NLOS signals. The clustering criteria obtained by offline dataset training are applied to detect multipath/NLOS signals for online data, which can enhance the performance of GNSS/INS real-time positioning. Additionally, it can provide an innovative perspective for the research on GNSS signal quality control methods of vehicle positioning systems in highly complex urban areas.

The rest of this paper is organized as follows: In Section 2, the relevant mathematical methodology is presented for K-means Clustering and GNSS/INS tightly coupled positioning algorithm. Next, Section 3 implements the data collection and experiment analysis, which validate the accuracy and reliability improvements of the proposed method. Finally, the conclusion and outlook will be given in Section 4.

## 2. Methodology

### 2.1. Feature Extraction

Reasonable feature value is critical to the capability of machine-learning algorithms, and this paper refers to the feature parameters of multipath/NLOS signal detection in supervised learning classifiers. Most of the current machine learning methods for multipath/NLOS detection adopt feature values at the observation data level. We only extract the feature parameters from the RINEX format file output by the GNSS receiver, including pseudorange, carrier phase, carrier-to-noise ratio (signal strength), and Doppler frequency shift, etc., which are all closely related to GNSS signal types. However, it is impossible for any single feature to effectively classify GNSS signals. Hence, a combination of different features is needed to ameliorate the classification accuracy [22,38,39,40].

(1) Satellite elevation angle: It is a common method to assign weights to each observation value based on the satellite elevation angle to reduce the influence of multipath and NLOS signal reception on the positioning results. Generally speaking, satellite signals from high elevation angles are less likely to be blocked and reflected by buildings, but this is not always the case in reality. Due to the height and distribution of buildings in urban areas, satellite signals at high elevation angles may also be NLOS signals, while signals at low elevation angles may be direct signals. Nonetheless, satellite elevation angle is still an important feature indicator to distinguish NLOS signals.

(2) Carrier-to-noise ratio: The GNSS receiver will output the observations of the tracked satellite signal strength. According to the signal propagation theory, supernumerary propagation and reflection will increase the path loss of the GNSS signal. As an important indicator reflecting the signal quality, the C/N0 observation value is also a common parameter to alleviate the multipath effect. Similar to the elevation angle, the satellite signal strength or carrier-to-noise ratio also has a certain correspondence with the type of signal. The signal strength of the satellite received by the survey antenna is usually higher in an open environment. However, the magnitude of C/N0 does not have a clear correspondence with the type of GNSS signal in a multipath environment, because constructive multipath will increase the received signal, while destructive multipath reduces signal strength.

(3) Pseudorange residual: When there are more observation equations than unknown parameters and the position estimation is accurate enough, the magnitude of the pseudorange residual can reflect the inconsistency between the pseudorange measurements and the geometric distance of the satellite.

In addition, multi-constellation GNSS integrated positioning increases the number of available observation satellites and observation redundancy. Therefore, the pseudorange residual can be used as an indicator to detect the quality of GNSS signals.

(4) Pseudorange rate consistency: The pseudorange observations originate from the receiver code tracking loop, and the Doppler shift of the signal is determined by the receiver frequency tracking loop. Compared with the code tracking loop, the multipath/NLOS signal has less influence on the frequency tracking loop, so the consistency between the pseudorange change rate and the Doppler frequency shift can reflect the interference degree of the reflected signal. Its formula is expressed as:(1)$$\zeta =|\Delta \rho -\overset{\text{˙}}{\rho }\cdot \Delta t|$$
where Δρ and Δt represent the pseudorange variation and time interval between adjacent epochs, respectively. According to the Doppler effect, the pseudorange rate $\overset{\text{˙}}{\rho }$ is calculated from the Doppler shift.
(2)$$\overset{\text{˙}}{\rho }=-\lambda_{i}\cdot f_{D_{i}}$$
where $\lambda_{i}$ and $f_{D_{i}}$ indicate the wavelength of frequency *i* and the Doppler shift in Hz, individually.

Since all of the above single features are uncertain and interdigitated with each other for NLOS signals, it is impossible for any single feature to effectively classify GNSS signals. Thus, NLOS signals need to be determined by a combination of different features. In summary, this paper comprehensively selects the above four parameters to form the feature vector of cluster analysis. Then the data are standardized to eliminate the influence of different dimensions on the clustering results, that is, each feature value conforms to the standard normal distribution after data processing.

### 2.2. K-Means Clustering Algorithm and Its Evaluation Indicator

It is considered that GNSS signals in complex environments are generally divided into two main types: direct signals and indirect signals (including multipath and NLOS signals), and each type of signal has a certain internal relationship with the above four feature parameters.

In accordance with this characteristic, the K-means algorithm is used for signal clustering. When the sample is closest to one of the cluster centers, it is classified into this class.

For a given sample set ${S=\text{\{}x}_{1}{,x}_{2},\cdots x_{m}\text{\}}$, where $x_{m}{=\text{\{}e}_{m}{,\mathrm{CN}0}_{m}{,v}_{m},\cdots {\mathrm{con}}_{m}\}$ represents the standardized feature vector of satellite elevation angle, carrier-to-noise ratio, pseudorange residual, and pseudorange rate consistency. This paper assigns corresponding weights to the feature parameters based on experience [22], which are set as $\left[\sqrt{0.2},\sqrt{0.3},\sqrt{0.2},\sqrt{0.3}\right]$, respectively. The K-means algorithm divides them into *k* clusters ${C=\text{\{}C}_{1}{,C}_{2},\cdots C_{k}\text{\}}$ so that the sum of squared Euclidean distances from each data point to its nearest cluster center is minimized, namely:(3)$${\mathrm{argmin}}_{C}\overset{k}{\underset{i=1}{\sum }}\underset{x\in C_{i}}{\sum }\|{x-\mu }_{i}\|{}^{2}$$
where $\mu_{i}$ is the mean vector of the cluster $C_{i}$, and can be expressed as:(4)$$\mu_{i}=\frac{1}{\left|C_{i}\right|}\underset{x\in C_{i}}{\sum }x$$

The basic process of the K-means algorithm is as follows [41]:

(1) This algorithm randomly selects *k* samples as the initial cluster center;

(2) For the remaining samples, according to the distance of their cluster centers, they are classified into the nearest cluster;

(3) For each cluster, the mean of all samples is calculated as the new cluster center;

(4) Repeat steps (2) and (3) until the cluster centers no longer change.

Based on the above calculations, all GNSS signals are classified into different clusters. The K-means algorithm needs to specify the value of *k* in advance, which is usually defaulted to 2 or 3. However, due to the complexity of the scenario and the correlation between GNSS signals, 2 and 3 are often not the optimal *k* values. Additionally, the difference between the clustering results corresponding to different *k* values is not obvious. Therefore, this paper chooses the Davies–Bouldin Indicator (DBI) as the internal evaluation index of the clustering effect [42].

DBI is defined as the average similarity between each cluster $C_{i}$, i=1,2,⋯,k and its the most similar one $C_{j}$, where the similarity is expressed by the ratio of the intra-cluster distance to the inter-cluster distance. The minimum value of DBI is 0, and the smaller the value, the better the clustering effect. The specific calculation formula is:(5)$$I_{\mathrm{DBI}}=\frac{1}{k}\overset{k}{\underset{i=1}{\sum }}{}_{\;i\neq j}^{\max}R_{\mathrm{ij}}$$
where *k* is the number of clusters; $R_{\mathrm{ij}}$ denotes similarity that can be constructed by a simple choice as follows so that it can keep nonnegative and symmetric:(6)$$R_{\mathrm{ij}}=\frac{d\left(s_{i}\right){+d(s}_{j})}{d_{\mathrm{ij}}}$$
where ${d(s}_{i})$ and ${d(s}_{j})$ mean the average distance between each point of cluster data to the centroid of that cluster also known as cluster diameter, individually; $d_{\mathrm{ij}}$ is the distance between cluster centroids *i* and *j*, which represents the dispersion degree of data point for cluster centroids *i* and *j*.

### 2.3. GNSS/INS Tightly Coupled Positioning Model

GNSS/INS integrated navigation system adopts Extended Kalman Filter (EKF) for system fusion to realize high-precision navigation and positioning by effectively detecting and rejecting Multipath/NLOS signals in complex urban areas. In vehicle navigation, due to the strong reliability and highly real-time performance of the pseudorange/INS system, this paper employs the integrated positioning solution of GNSS double-differenced pseudorange (DGNSS) and INS observation.

The system state model depends on the INS error model and the description of the inertial sensor system error. The INS error equation based on the psi angle is adopted in this paper [43].
(7)$$\delta {r\text{˙}}^{n}{=\;-\omega }_{\mathrm{en}}^{n}{\text{×}\delta r}^{n}{+\delta v}^{n}$$
(8)$$\delta {v\text{˙}}^{n}{=\;-(2\omega }_{\mathrm{ie}}^{n}{+\omega }_{\mathrm{en}}^{n}{)\text{×}\delta v}^{n}{-\psi }^{n}{\text{×}f}^{n}{+\delta g}^{n}{+C}_{b}^{n}{\delta f}_{\mathrm{ib}}^{b}$$
(9)$${\overset{\text{˙}}{\psi }}^{n}=\;-\left(\omega_{\mathrm{ie}}^{n}{+\omega }_{\mathrm{en}}^{n}\right){\text{×}\psi }^{n}{-C}_{b}^{n}{\delta \omega }_{\mathrm{ib}}^{b}$$
where *δ*$r^{n}$*, δ*$v^{n}$*,* and *δ*$\psi^{n}$ indicate position error, velocity error, and attitude angle error, respectively; $C_{b}^{n}$ is the rotation matrix from the body frame (*b*-frame) to the navigation frame (*n*-frame); ${\delta f}_{\mathrm{ib}}^{b}$ and ${\delta \omega }_{\mathrm{ib}}^{b}$ are the accelerometer and gyroscope error vector in the *b*-frame, separately; In addition, the specific force vector measured by the accelerometer, the rotation velocity of the earth, and the transfer rate are represented by $f^{n}$, $\omega_{\mathrm{ie}}^{n}$, and $\omega_{\mathrm{en}}^{n}$, respectively. 

The accelerometer error and gyroscope error are the main factors affecting the accuracy of GNSS/INS tightly coupled system, and the bias errors are modeled by a random walk process. Their specific forms can be expressed as:(10)$$\left\{\begin{array}{c} {\overset{\text{˙}}{b}}_{a}{=w}_{\mathrm{ba}}\\ {\overset{\text{˙}}{b}}_{g}\;{=w}_{\mathrm{bg}} \end{array}\right.$$
where $b_{a}$ is bias of the accelerometer; $b_{g}$ is the bias of the gyroscope; $w_{\mathrm{ba}}$ and $w_{\mathrm{bg}}$ express the corresponding random white noise.

The equation of the state of the system is as follows:(11)$${\overset{\text{˙}}{X}}_{\mathrm{ins}}=F\cdot X_{\mathrm{ins}}+G\cdot W$$
where F is the state transition matrix; G and W represent the dynamic noise matrix and the noise vector; $X_{\mathrm{ins}}$ is the state parameter, $X_{\mathrm{ins}}{=(\delta r,\delta v,\delta \psi ,bg,ba)}_{15\text{×}1}$. 

Based on the INS error model, G and W are optimized and presented as follows:(12)$$F=\left[\begin{array}{ccccc} F_{\mathrm{rr}} & F_{\mathrm{rv}} & 0 & 0 & 0\\ F_{\mathrm{vr}} & F_{\mathrm{vv}} & f^{n}\text{×} & 0 & C_{b}^{n}\\ F_{\psi r} & F_{r\psi } & -\left(\omega_{\mathrm{ie}}^{n}{+\omega }_{\mathrm{en}}^{n}\right)\text{×} & -C_{b}^{n} & 0\\ 0 & 0 & 0 & 0 & 0\\ 0 & 0 & 0 & 0 & 0 \end{array}\right],G=\;\left[\begin{array}{ccccc} 0 & 0 & 0 & 0 & 0\\ 0 & C_{b}^{n} & 0 & 0 & 0\\ {-C}_{b}^{n} & 0 & 0 & 0 & 0\\ 0 & 0 & 0 & I & 0\\ 0 & 0 & 0 & 0 & I \end{array}\right]$$
where I indicates the unit matrix; $F_{\mathrm{rr}}$ and $F_{\mathrm{rv}}$ represent state coefficients of position; $F_{\mathrm{vr}}\;$ and $F_{\mathrm{vv}}$ represent coefficients of velocity; $F_{\psi r}$ and $F_{r\psi }$ represent coefficients of attitude. The specific derivation of the above symbols can be found in [33].

The difference between the distance from the satellite to the ground station predicted by the INS and the GNSS double-differenced pseudorange is solved, which is used as the EKF measurement to achieve high-precision positioning for GNSS/INS tightly coupled system. The measurement equation is written in matrix form:(13)$$Z_{k}{=H}_{k}X_{k}+{\left[e_{\rho }{,e}_{\mathrm{ins}}\right]}^{T}$$
where $Z_{k}$ represents the measurement vector at time epoch *k*, $Z_{k}=\Delta \nabla \rho^{*}-\Delta \nabla \rho_{\mathrm{ins}}$, and $\Delta \nabla \rho_{\mathrm{ins}}$ indicates the satellite-to-ground distance predicted by INS; “*” represents different satellite systems including uses GPS and BDS in this paper; $H_{k}$ is the measurement model coefficient matrix; $e_{\rho }$ and $e_{\mathrm{ins}}$ are the pseudorange observation noise and INS observation noise, respectively.

The final GNSS/INS tightly coupled positioning results can then be solved based on the following EKF procedures.

Prediction stage:(14)$${\hat{X}}_{k/k-1}{=\Phi }_{k,k-1}{\hat{X}}_{k-1}$$
(15)$$P_{k/k-1}{=\Phi }_{k,k-1}P_{k-1}\Phi_{k,k-1}^{T}{+Q}_{k-1}$$

Update stage:(16)$$K_{k}{=P}_{k/k-1}H_{k}^{T}{\left(H_{k}P_{k/k-1}H_{k}^{T}{+R}_{k}\right)}^{-1}$$
(17)$${\hat{X}}_{k}={\hat{X}}_{k/k-1}{+K}_{k}\left(Z_{k}\;{-H}_{k}{\hat{X}}_{k/k-1}\right)$$
(18)$$P_{k}=\left({I-K}_{k}H_{k}\right)P_{k/k-1}$$
where ${\hat{X}}_{k}$, $\;\Phi_{k}$, and $P_{k}$ express the state vector estimates, the state transition matrix, and the error covariance matrix at time epoch *k*, respectively; $Q_{k-1}$ represents the system noise covariance matrix at time epoch *k*−1; $R_{k}$ indicates the measurement noise covariance matrix at time epoch *k*; $H_{k}$ denotes the measurement matrix at time epoch *k*; $K_{k}$ represents the EKF gain matrix at time epoch *k*; In addition, ∎*_k_*_,*k*−1_ represents matrix/vector ∎propagation from time epoch *k*−1 to *k*.

### 2.4. Overview of the Proposed Method

The flowchart of the proposed method is shown in Figure 1. Firstly, four essential features are extracted from GNSS raw observation data, namely, satellite elevation angle, carrier-to-noise ratio, pseudorange residual, and pseudorange rate consistency, which are comprehensively used to enhance the classification accuracy. Secondly, satellite signals received by GNSS receivers in complex scenarios are generally divided into two categories: direct and indirect signals, the latter including multipath and NLOS signals, and each type of signal has a certain internal relationship with the above four main features. According to this characteristic, this paper adopts the k-means clustering algorithm to cluster the signals and selects the DBI as the internal evaluation index of the clustering effect. Finally, both the raw INS measurements and the GNSS double-differenced pseudorange observations are tightly integrated with EKF filtering, resulting in reliable and high-precision positioning results.

It should be pointed out that the measured circumstance of the GNSS receiver of the base station is open and superior with no obstacle occluding all around as shown in Figure 2. Additionally, the Trimble GNSS-Ti earth-type chock-ring antenna is installed. This paper purports that there is no NLOS signal and the multipath is well suppressed, which is small relative to the rover and does not affect the subsequent positioning solution. Therefore, multipath/NLOS signal detection for base station GNSS observations is not required.

## 3. Data Collection and Experiment Analysis

### 3.1. Data Collection

This paper takes advantage of GPS L1 and BDS B1 frequency observation data, and the sampling rate is 1 Hz. The reference values of the three-dimensional position, velocity, and attitude of the vehicle are acquired through GNSS RTK/INS tightly coupled positioning, which is used to verify the improvement of positioning accuracy after excluding outliers. The NovAtel ProPak6 receiver of the rover station and the Trimble receiver of the three static base stations provide the GNSS carrier phase observations. Moreover, the solution of the reference value is realized using Inertial Explorer post-processing software developed by NovAtel Company [44]. The experimental platform is shown in Figure 3, and Table 1 is the specific technical parameters of the IMU.

The experiment was carried out in the urban area of Nanjing, where the typical scenarios included tree shade and urban canyons, etc. Figure 4 shows the vehicle trajectories corresponding to two different observation periods. The two sets of datasets are marked as D1 and D2 in the order of time. D1 is an offline dataset, and D2 is an online dataset for real-time positioning verification. In addition to the occlusion of satellite signals caused by obstacles such as huge buildings, the strong reflection effect of modern building materials can also cause serious multipath and NLOS signal reception. Furthermore, the lush tree canopy on both sides of urban roads can also lead to complex multipath effects.

The specific information of the dataset is displayed in Table 2. The number of valid epochs refers to the number of epochs for which the position solution is obtained. Owing to the influence of the observation environment, some epochs do not have GNSS data output, or the number of observation satellites is too small to conduct the double system positioning solution. Therefore, these epochs will be considered to be invalid epochs.

### 3.2. Outliers Detection for the Offline Data Based on K-Means

This section will discuss in detail how to use the K-means algorithm to detect GPS/BDS multipath/NLOS signals on dataset D1, which employs GNSS/INS tightly coupled post-processing algorithm to verify its effectiveness. This lays the foundation for the subsequent real-time application for GNSS/INS integrated navigation.

In order to broaden the feature value range of the sample and upgrade the performance of the machine learning algorithm, the cut-off elevation angle and the signal-to-noise ratio are not set during the positioning process. The number of epochs that do not satisfy the chi-square test accounts for about 10% of the valid epochs, indicating that the observation environment has a certain complexity. The epochs that satisfy the chi-square test still contain a certain number of satellite observations disturbed by multipath and NLOS signals. Therefore, identifying them accurately is the key to enhancing the performance of GNSS positioning and GNSS/INS tightly coupled positioning. Figure 5 illustrates the sky map of GPS and BDS observed by the receiver during vehicle driving.

#### 3.2.1. Cluster Analysis

Since only four typical feature parameters are used as the input of the machine learning model, there is no need to use methods such as Principal Components Analysis (PCA) for dimension reduction of the sample data. The feature value is normalized by a z-score to ensure that they are in the same order of magnitude. Here, different weights are assigned to the four feature parameters elaborated above for clustering calculation. Due to the possible connectivity and correlation between different GNSS signal types and the potential non-integrity of feature parameters, we will no longer constrain the value of *k* to 3 or 2, but judge according to the actual situation.

Figure 6 demonstrates the corresponding DBI values for different values of *k*. It can be revealed that when the value of *k* is 3, 4, 5, and 7, the DBI value is small with inconspicuous disparity. Hence, this paper selects the number of clusters from these four groups of values.

It is worth noting that due to the incompleteness of feature parameters, the complexity of environmental influences and the fuzzy correlation between signal types, the number of clusters do not necessarily correspond strictly to the type of signal (three types of LOS/Multipath/NLOS); that is, 3 is not necessarily the optimal value of *k*. Here only the value of DBI is used as a reference for us to determine the value of *k*. *k* = 3 is great, but if the DBI value of *k* > 3 is also low, we will also consider it. In this case, we can think of them as clusters with varying degrees of signal interference. Due to the existence of inertial measurement data in tightly combined positioning, we do not need to care too much about the perfect value of *k* as in GNSS alone positioning for the time being. A slightly larger division of GNSS signals into multipath /NLOS will not cause the underdetermination of the tightly combined observation equation.

Figure 7 shows the visualization graphics of the clustering results corresponding to different *k* values in the three-dimensional feature space. In general, the larger the value of *k*, the smaller the number of observations in the dataset that are not disturbed by multipath and NLOS signals. For example, when *k* is 3, we know from observation in Figure 7, and from experience, that the number of line-of-sight (LOS) observations is 7807, and the number of samples of the other two clusters is 3646 and 1. When *k* is 7, the number of LOS observations is reduced to 6563.

Since the larger *k* is, the smaller is the number of LOS observations, if only the LOS observations are retained during the positioning process, the risk of *k* = 3 lies in missed detection, and the false detection rate will increase when *k* = 7. Considering that there is no extraordinary explicit boundary between the observation data of different characteristics, this paper chooses a relatively compromised method that the value of *k* is set to 5. This way can alleviate the probability that observations are contaminated by multipath and NLOS, which are mixed into the LOS clusters during the clustering process; thus, it can avoid a reduction in the accuracy of GNSS/INS tightly coupled positioning. Furthermore, based on the existence of INS observation data, positioning results can be output, despite an insufficient number of LOS satellites. At this time, the number of samples of LOS observations is 7286.

Figure 8 further expresses the distribution pattern of the clustering results under four different feature parameters when *k* = 5. The diagonal represents the probability distribution of the sample points of different clusters on the variable. Additionally, the off diagonal represents the horizontal distribution of the sample points on the corresponding two-dimensional feature. 

It is worth noting that here for better visualization, the probability-density curves of each cluster are independently normalized so that their respective acreage under each curve is 1. According to the probability distribution, it can be inferred that cluster 1 in Figure 8 is the LOS observation sample. It can be seen that its main distribution area is concentrated in the space with the maximum satellite elevation angle and carrier-to-noise ratio, the smallest pseudorange residual, and the consistency of the pseudorange rate, in which the elevation angle of the satellites is basically above 25°. This is highly similar to the nature of LOS observations in urban environments. The remaining four clusters can be considered to be different distribution patterns of the sample set composed of observations contaminated by multipath or NLOS signals.

#### 3.2.2. Analysis of Positioning Results

In order to validate the correctness of the clustering results, we execute the DGNSS/INS tightly coupled positioning algorithm to conduct comparative experiments. Figure 9 demonstrates the comparison of the 3D position of the vehicle obtained by different positioning methods, where the red solid line is the ground reference value of the vehicle on the ground. This paper takes the starting point of the vehicle as the origin of the site-centric coordinate system. From Figure 9, we can recognize that the tightly coupled positioning result without multipath/NLOS observation processing deviates more from the reference coordinate value. Furthermore, the result without the height cut-off elevation angle has the most serious position deviation. The deviation in the starting position from the reference value causes an adverse influence on the positioning accuracy. 

Since the multipath/NLOS observations are not eliminated in complex urban scenarios, it makes the tightly coupled observation equation unreliable. It can also be concluded that installing the satellite cut-off elevation angle can suppress the influence of multipath/NLOS observations to a certain extent. However, the overall positioning performance of the GNSS/INS tightly coupled system is still unsatisfactory, especially in the vertical direction. However, after using the K-means clustering algorithm to identify and exclude multipath/NLOS observations, the positioning results are significantly ameliorated, and the deviation level from the true value becomes small. 

Moreover, even if the satellite cut-off elevation angle is set to 0°, the position solution of this method hardly changes. This phenomenon indicates that the cut-off elevation angle of most LOS observations in the scenario identified by the clustering algorithm are above 15°. However, it does not mean that all satellite observations above 15° come from LOS signal reception.

The above experiments also reveal that simply relying on setting the cut-off elevation angle cannot improve the positioning accuracy of GNSS/INS tightly coupled system, and even destroy its satellite geometry, which is proved and expounded in detail by Xia et al. (2020). The LOS signal identification method proposed in this paper sufficiently considers the influence of the four feature parameters on the properties of GNSS signals with higher reliability.

Furthermore, this paper calculates the RMS of the position error for the above positioning method, as shown in Table 3. The experimental results show that the positioning accuracy is 0.63 m in the horizontal direction and as high as 6.50 m in the vertical direction without multipath/NLOS observation processing (the cut-off elevation angle is set to 15°).

After excluding the multipath/NLOS observations, the GNSS/INS tightly coupled positioning accuracy is improved by 16% and 85% in the horizontal and vertical directions, respectively. The horizontal accuracy reaches 0.53 m, and the vertical accuracy is specially promoted to within 1 m. It is particularly acknowledged that if using the K-means algorithm to eliminate multipath and NLOS observations, we do not need to set the cut-off elevation angle, because the algorithm has taken into account the factor that the low elevation angle is susceptible to multipath/NLOS signal interference. Additionally, compared to the accuracy of the positioning result in the horizontal direction, the accuracy in the vertical direction tends to be substantially promoted as the multipath/NLOS signals are eliminated. This is because in the harsh urban scenarios, the positioning results in the vertical direction are more severely affected by multipath/NLOS signals than the positioning results in the horizontal direction, which is elaborated upon by Sun et al. (2022).

To more clearly show the contribution of K-means clustering to the positioning results, we implemented GNSS pseudorange double-differenced positioning for the experiments. Figure 10 shows the comparison of positioning trajectories in several typical real urban scenarios (skyscrapers and trees), which demonstrates that the results of GNSS/INS tightly coupled position are significantly better than GNSS. Further, the accuracy of GNSS pseudorange differential positioning has been greatly promoted based on K-means to exclude multipath/NLOS observations. With the addition of INS observation data, not only the accuracy of the positioning consequence is further improved, but also possesses better continuity and availability.

Figure 11 and Figure 12 show the comparison of the number of satellites participating in the position calculation and the PDOP value before and after excluding the multipath/NLOS observations, respectively. By comparison, it is known that after excluding satellites, the number of effective satellites is expected to decrease significantly, which causes the geometric distribution of satellites to be more deteriorated. This phenomenon indicates that the experimental environment is relatively harsh, and multipath/NLOS satellites account for a considerable proportion; yet, despite this, the positioning accuracy has been greatly ameliorated, which shows that the multipath/NLOS signal detection algorithm in this paper is effective. 

### 3.3. Outliers Detection for Real-Time Positioning

We use offline data for unsupervised learning and training to obtain the signal type identification rule. This rule is finally used for real-time GNSS observation classification and position calculation, and we call the real-time observed data—online data. Due to the good scalability of the K-means algorithm, there is no need for re-supervised learning on offline data (the labels have been obtained through the above process), but the clustering rules obtained by training are directly used for real-time detection of signals. Therefore, the above model is then employed to identify the observation outliers containing multipath/NLOS signal on the online observation dataset D2 for GNSS/INS real-time positioning. Feeding new GNSS/INS observations into the classifier, outliers can be obtained in real time. Similar to the offline system, the accuracy and availability of INS/DGNSS double-differenced pseudorange positioning results are used to evaluate the performance of anomaly detection algorithms after excluding multipath/NLOS signals.

Figure 13 indicates the comparison of the number of satellites participating in the position calculation before and after excluding the multipath/NLOS observations.

Figure 14 exhibits the comparison of the 3D position of the vehicle obtained by different positioning methods, where the red solid line is the ground truth value of the vehicle. From Figure 4b, Figure 13 and Figure 14, we can see that the observation conditions of the first part of the epoch of the data set D2 are poor. Multipath/NLOS signals seriously affect the positioning accuracy, which causes the position result to deviate from the reference value to a large extent. Without detecting and eliminating multipath/NLOS on the observational data, the trajectory diagram of the vehicle’s 3D position can even produce spurs. The observation circumstance of the subsequent epochs is gradually expanded, and the results obtained by the two algorithms are close to the reference value. This demonstrates that in urban complex scenarios, detecting and eliminating multipath/NLOS signals plays an important role in improving the performance of GNSS/INS integrated real-time positioning.

Further, this paper calculates the RMS value of the position error of the real-time positioning of the online dataset, as shown in Table 4. It can be seen that in the urban harsh scenarios, the K-means clustering algorithm has a better effect on the improvement of the positioning results. The positioning accuracy of the D2 online dataset is improved by 21% and 41% in the horizontal and vertical directions, respectively.

## 4. Discussion

Multipath effects and NLOS signals are the main factors restricting the accuracy and reliability of GNSS/INS positioning, especially in challenging environments such as urban canyons, shaded trees, etc. Therefore, given the interference of multipath/NLOS signals, this paper proposes an outliers detection method composed of an offline learning system and an online learning system for GNSS/INS tightly coupled positioning in urban areas. We believe that GNSS signals in complex environments are generally divided into three categories: LOS, multipath, and NLOS signals. Each type of signal has a certain internal relationship with the four feature parameters of satellite elevation angle, carrier-to-noise ratio, pseudorange residual, and pseudorange rate consistency. According to this characteristic, K-means algorithm is used for signal clustering. When the sample is closest to one of the cluster centers, it is classified into this class. In an offline system, the K-means clustering algorithm is employed to detect observation outliers and construct an offline training set with labels, without resorting to 3D building model and external sensor. On this basis, due to the good scalability of the K-means clustering algorithm, the above-mentioned model is then utilized to identify multipath/NLOS signals on the online observation dataset for real-time positioning.

As can be seen from Figure 11 and Figure 12, after excluding GNSS observation outliers, the number of satellites participating in the position calculation decreases, and the GDOP value has generally increased. Yet, despite this, the positioning accuracy has been improved. However, while ensuring the positioning accuracy, the continuity of the dynamic positioning results is also crucial. Although correct outlier detection and exclusion can effectively improve the performance of positioning results, it must be admitted that directly removing abnormal observations will reduce the number of available GNSS satellites and weaken the satellite geometric distribution. This will reduce the positioning performance to a certain extent, especially when the number of available satellites is small. Therefore, it is not advisable to blindly pursue positioning accuracy and lose a large number of original valid epochs, but there will certainly be more space for optimizing signal selection when the available GNSS constellations are enough.

Unfortunately, although this paper has made some meaningful explorations in multipath/NLOS detection and elimination, the above research work still needs to be further improved because of the complexity of multipath/NLOS signals, for example, using more GNSS/INS observation data to establish offline label datasets, so that the training set covers more scenarios and satellite constellations, and improves the generalization ability of the classification model. In addition, based on different anomaly distribution assumptions, a more suitable detection method for observation outliers is pursued under the condition of ensuring positioning accuracy. We have also made preparations for this and will further study it in the future.

## 5. Conclusions

GNSS/INS integrated navigation possesses excellent characteristics so that it plays a significant role in vehicle positioning requirements. However, the performance of GNSS/INS integration suffers from excessive unexpected GNSS outliers such as multipath/NLOS signal in dense urban areas. 

This paper put forward an urban vehicle GNSS multipath/NLOS observation detection algorithm based on K-means clustering, which can effectively promote the accuracy of GNSS/INS tightly coupled positioning results. The method is essentially an offline learning system that can be used for post-processing solution of GNSS/INS observation data. Simultaneously, we employ K-means to detect observation outliers and obtain LOS/NLOS classification rules, which can be further broadened to GNSS/INS integrated navigation vehicle position in real-time. The proposed method obtains the signal type label by sufficiently excavating the information of the GNSS observation data itself, without the assistance of external software and hardware. Based on the good scalability of the K-means clustering algorithm, the above model is used to identify the multipath/NLOS of online observation data for real-time positioning. As a result, it can effectively enhance the performance of GNSS/INS tightly coupled system with higher availability and environmental adaptability.

In future work, we will continue to research the influence of the GNSS signal distribution pattern in different scenarios and test data on the positioning performance of GNSS/INS tightly coupled system, and study a more robust outlier boundary determination rule. Additionally, when the number of visible satellites is relatively small, simply excluding the multipath/NLOS signal will deteriorate the satellite geometric distribution, which reduces the positioning accuracy or even fails to execute the positioning solution. Therefore, in future research, we will further consider reasonable multipath/NLOS processing strategies, such as optimizing the stochastic model of the observation equation [45].

## Figures and Tables

**Figure 1 micromachines-13-01128-f001:**
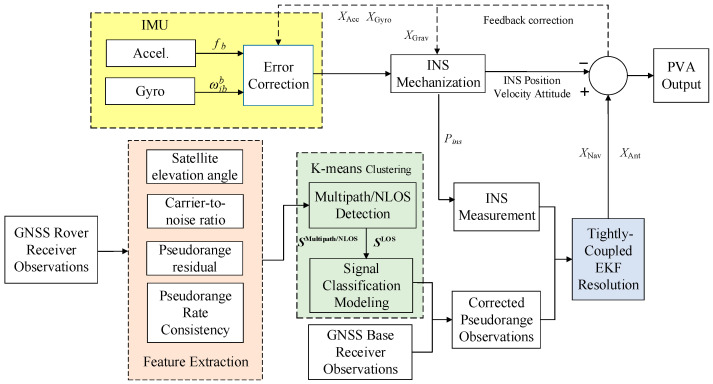
Flowchart of the proposed Multipath/NLOS Detection method based on K-means Clustering for GNSS/INS tightly coupled system.

**Figure 2 micromachines-13-01128-f002:**
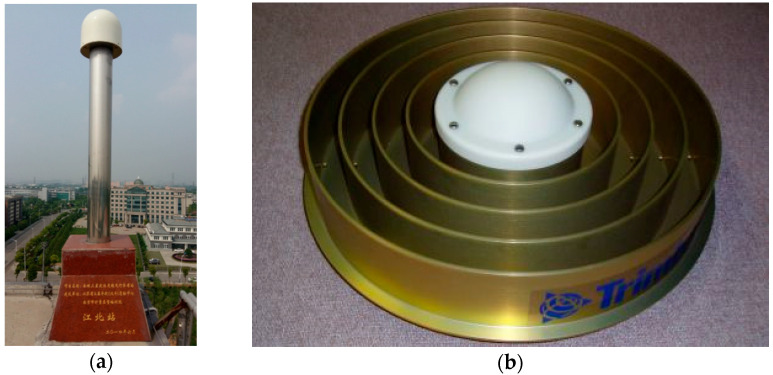
Observation environment of the base station and GNSS receiver: (**a**) The base station; and (**b**) Trimble GNSS-Ti earth-type chock-ring antenna.

**Figure 3 micromachines-13-01128-f003:**
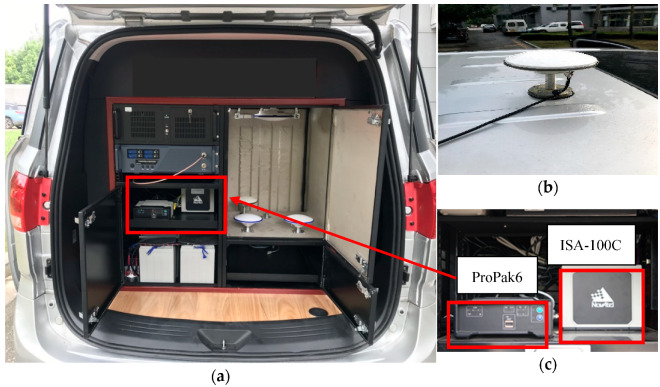
The test platform and equipment of the experimental vehicle: (**a**) The measurement vehicle; (**b**) satellite signal receiving antenna; (**c**) integrated system sensors.

**Figure 4 micromachines-13-01128-f004:**
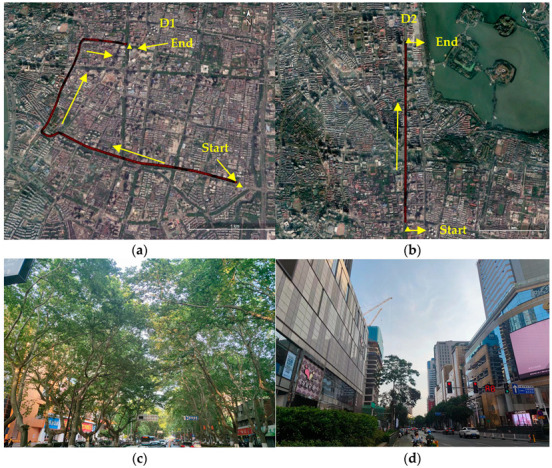
The route map of the actual measurement of the urban complex scenarios: (**a**) the route maps of offline dataset D1 for vehicle dynamic testing; (**b**) the route maps of online dataset D2 for vehicle dynamic testing; (**c**) the boulevard; (**d**) the urban canyon.

**Figure 5 micromachines-13-01128-f005:**
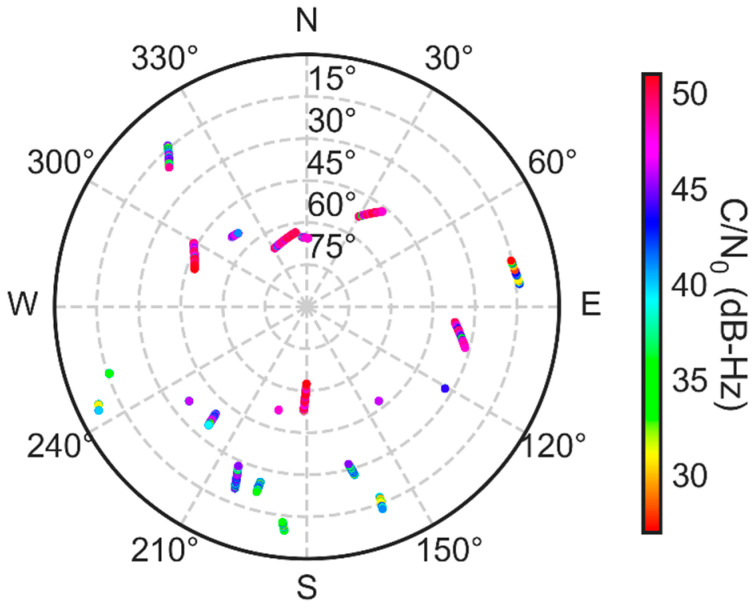
GPS/BDS sky map and observation signal-to-noise ratio.

**Figure 6 micromachines-13-01128-f006:**
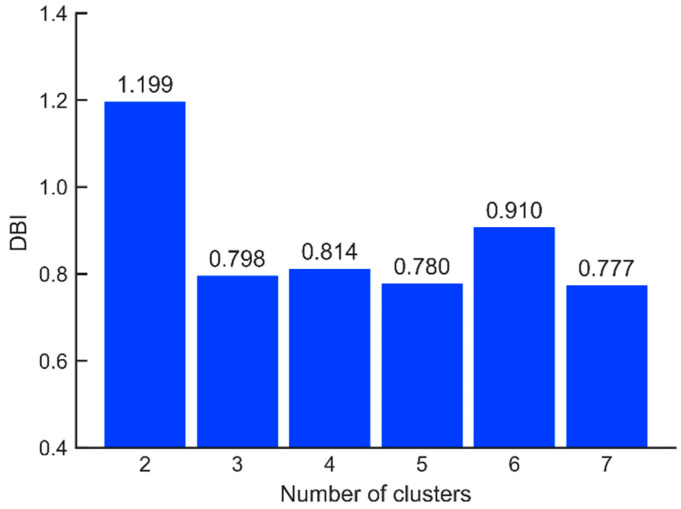
DBI values of clustering results correspond to different *k* values.

**Figure 7 micromachines-13-01128-f007:**
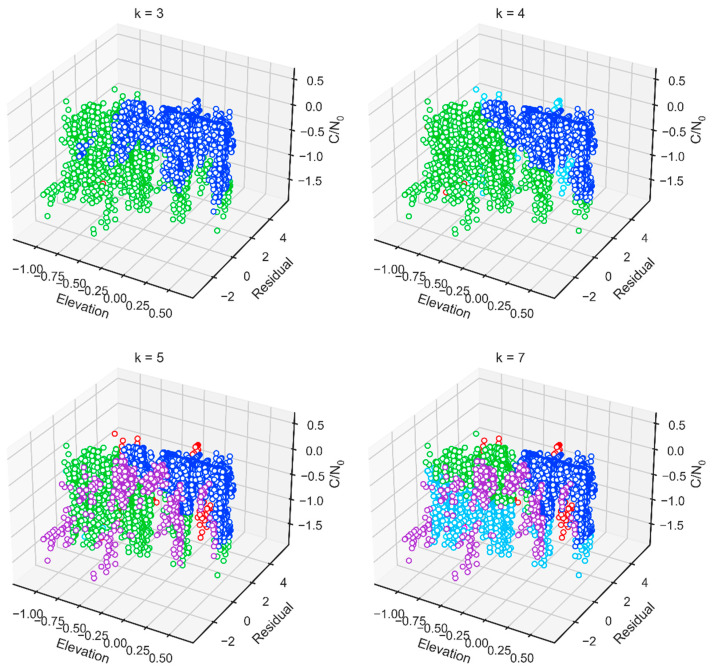
Clustering effect illustration based on different *k* values.

**Figure 8 micromachines-13-01128-f008:**
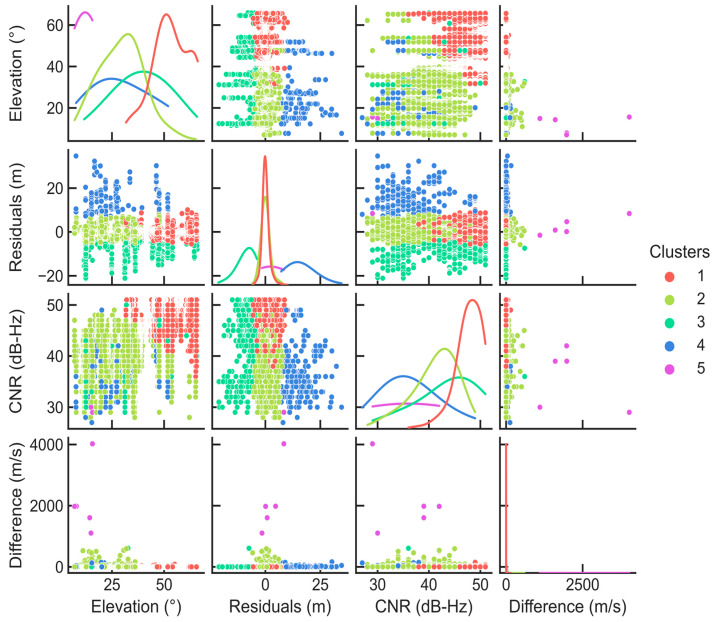
Four feature value distribution pattern diagrams when the *k* value is 5.

**Figure 9 micromachines-13-01128-f009:**
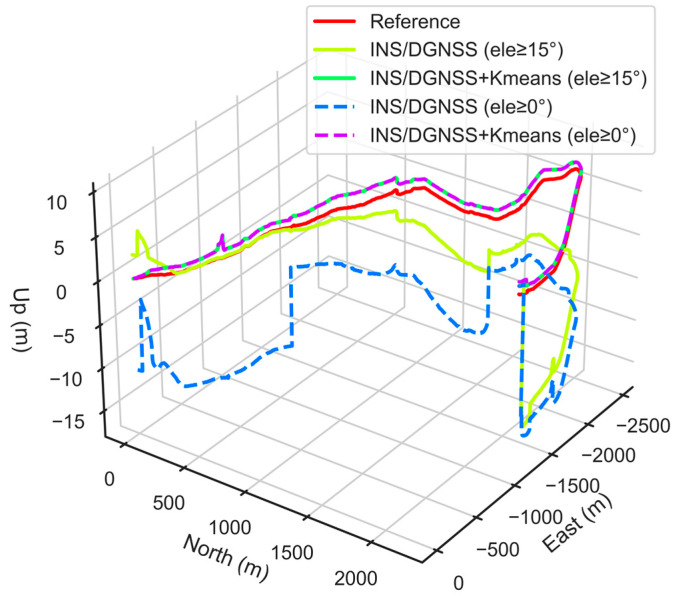
3D position comparison of positioning results of different tightly coupled schemes.

**Figure 10 micromachines-13-01128-f010:**
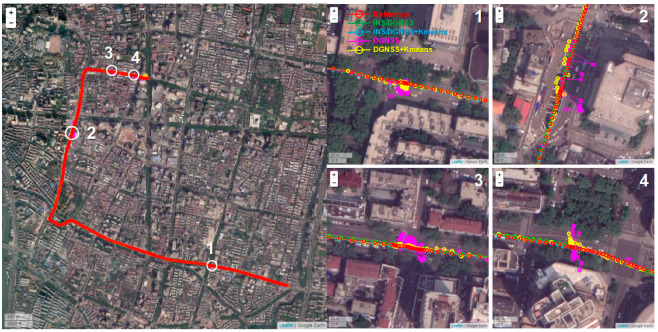
Comparison of positioning results in typical occlusion environments.

**Figure 11 micromachines-13-01128-f011:**
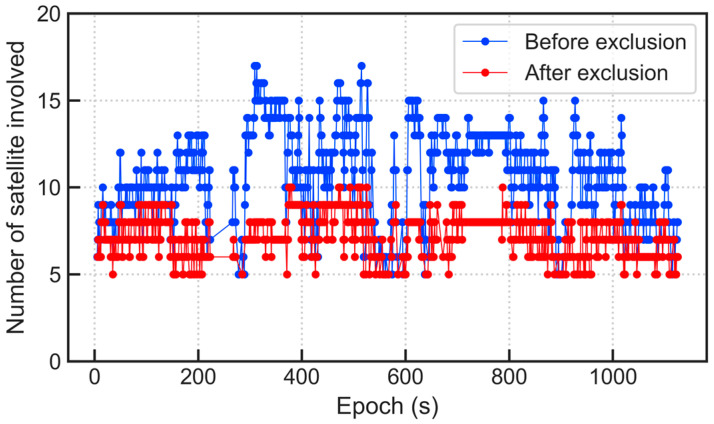
Comparison of the number of satellites before and after excluding multipath/NLOS signals.

**Figure 12 micromachines-13-01128-f012:**
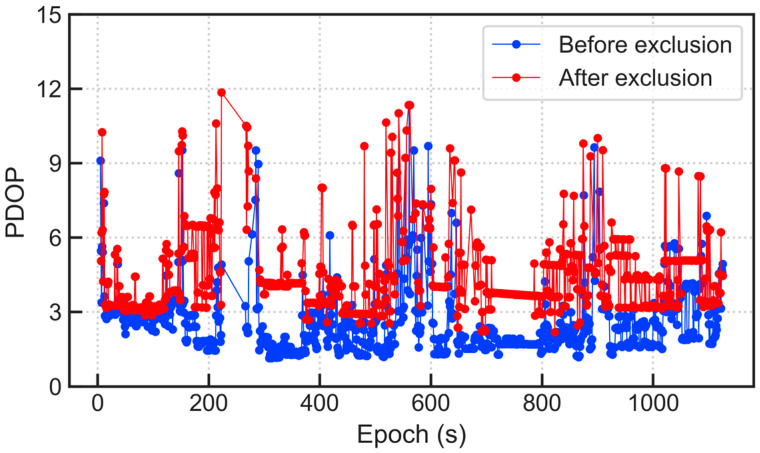
Comparison of PDOP values before and after excluding multipath/NLOS signals.

**Figure 13 micromachines-13-01128-f013:**
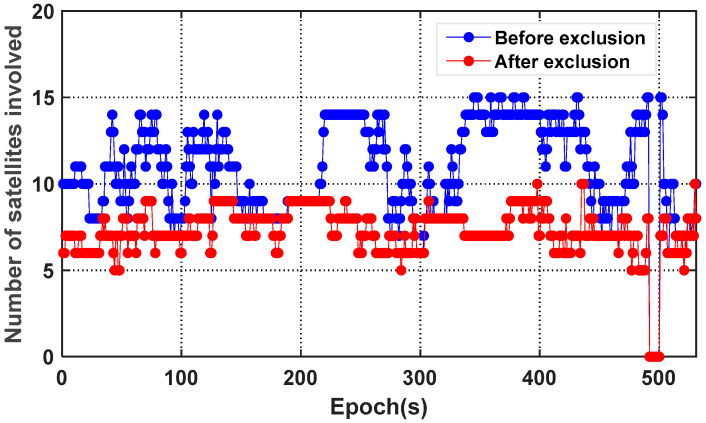
Comparison of the number of satellites for D2 before and after excluding multipath/NLOS signals.

**Figure 14 micromachines-13-01128-f014:**
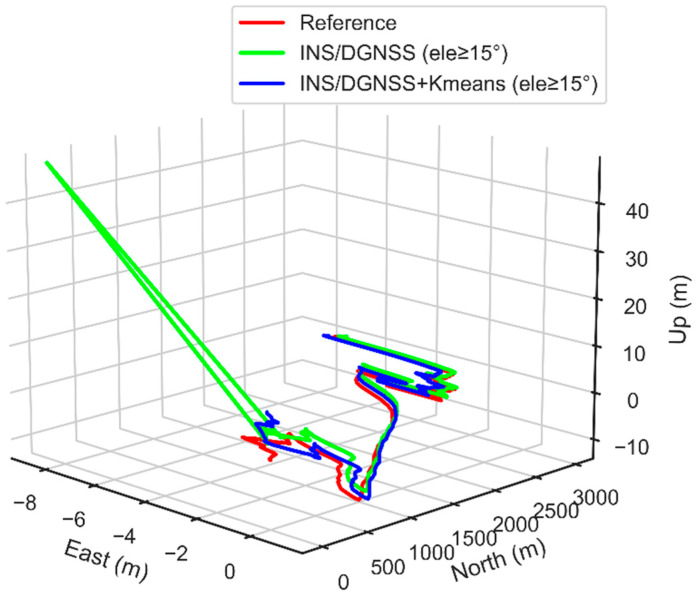
3D position comparison of positioning results of different tightly coupled schemes.

**Table 1 micromachines-13-01128-t001:** IMU technical data.

Parameters	Gyroscope	Accelerometer
Bias stability	0.5 deg/h	1250 μg
Scale factor	100 ppm	100 ppm
Random walk	$0.3\deg/\sqrt{h}$	100 μg $/\sqrt{h}$

**Table 2 micromachines-13-01128-t002:** Valid epoch and sample size of each dataset.

Dataset	Start Time (UTC)	End Time (UTC)	Valid Epoch	Sample Size
D 1	2017-04-20 05:12:58	2017-04-20 05:31:43	1052	11,454
D 2	2017-04-20 05:31:44	2017-04-20 05:41:28	531	6011

**Table 3 micromachines-13-01128-t003:** Accuracy comparison of positioning results for different tightly coupled schemes.

Positioning Scheme	RMS/m			
	E	N	U	2D
INS/DGNSS (ele ≥ 15°)	0.33	0.53	6.50	0.63
INS/DGNSS (ele ≥ 0°)	0.32	2.75	11.57	2.76
INS/DGNSS + K-means (ele ≥ 15°)	0.37	0.38	0.98	0.53
INS/DGNSS + K-means (ele ≥ 0°)	0.38	0.38	0.98	0.53

**Table 4 micromachines-13-01128-t004:** Accuracy comparison of positioning results for different tightly coupled schemes.

Positioning Scheme	RMS/m			
	E	N	U	2D
INS/DGNSS (ele ≥ 15°)	0.53	0.54	2.79	0.75
INS/DGNSS + K-means (ele ≥ 15°)	0.53	0.25	1.66	0.59

## Data Availability

The data presented in this study are available on request from the corresponding author.

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
