# Peer review of "Multipath/NLOS Detection Based on K-Means Clustering for GNSS/INS Tightly Coupled System in Urban Areas"

_micromachines, 2022, doi:10.3390/mi13071128_

Round 1
Reviewer 1 Report
The manuscript proposed K-means clustering algorithm for GNSS/INS NLOS and multipath detection. The strategy of the algorithm has been well stated and the two field tests has been carried out to verify the proposed algorithm. The following listed comments needs to be clarified.
1. Line 292, the two field tests are distinguished the datasets to D1 and D2, it could be easy understanding if the D1 and D2 are marked on the corresponding plots on Figure 4.
2. Line 189 to 296, the complexed test environment is described in details. It would be helpful to plot the overall visible satellites along with the environment description.
3. The study uses the DBI to determine the cluster index, and the number of clusters with respect to. DBI values is plotted. It would be helpful to further discuss the relationship between clusters number and DBI values.
4. Line 337 to line 351, this section discussed the results of different clusters number
5. In this section, the clustering results of different k are discussed and number of LOS observations are given. It would be helpful to explain in detail about the process of determined the LOS observation from different clustering results. According to the manuscript, the number of the LOS observation changes according to the different clustering number, the effectiveness of the algorithm should also be studied.
6. The study set the number of k to 5. It would be helpful to present more details about this determination. Besides LOS, NLOS and multipath, which are the other two groups of observations?
7. In figure 8, The coordinate axes of diagonal plots are confusing. As the manuscript states that the diagonal plots are probability distribution of different clusters. the horizontal axis and vertical axis of the diagonal plots, however, are the same units, which is not rigorous to represent the probability distributions.
8. Line 378 and 379 states that the trajectory of GNSS/INS tightly integrated deviated from the reference trajectory. The bias of the 3D trajectory of GNSS and INS, however, exists a systematic bias in UP component, which may not cause by the NOLS or multipath error. It would be helpful to analysis the ENU components error individually.
9. In line 383, the manuscript concludes that the tightly coupled integration model is unreliable, even using an adaptive algorithm. I couldn’t find any numerical analysis to support the latter conclusion. The main purpose of adaptive algorithm is adjusting the contribution of the dynamic model on the state estimation, the algorithm is hardly affecting the influences of the poor observations. It would be helpful to given the conclusion after the analysis.
10. The manuscript carried out two field tests to test the effectiveness of the proposed algorithm, one is offline training and another one is online training. However, I couldn’t find the strategy differences between the online and offline training. In addition, the error on up components of the two data sets are different, it would be helpful to analysis the reason of the differences.
11. the authors surnames of Reference 22 are incorrect.
Reviewer 2 Report
Multipath/NLOS Detection based on K-means Clustering for GNSS/INS tightly coupled system in Urban Areas
The accuracy and reliability of GNSS positioning can be severely degraded in a highly urbanized area because of multipath effects and non-line-of-sight (NLOS) signal receptions. In this paper, the authors propose a multipath/NLOS detection method based on K-means clustering algorithm for vehicle GNSS/INS integrated positioning. It comprehensively considers different feature parameters derived from GNSS raw observations, such as satellite elevation angle, carrier-to-noise ratio, pseudo-range residual, and pseudo range rate consistency to effectively classify GNSS signals. The K-means clustering algorithm is exploited to divide the observation data into two main categories: direct signals and indirect signals. Several experiments conducted in typical urban scenarios demonstrate that the proposed method could ameliorate the positioning accuracy significantly compared with the conventional GNSS/INS integrated navigation. After removing GNSS outliers, the positioning accuracy of the offline dataset is improved by 16% and 85% in the horizontal and vertical directions, respectively, and the positioning accuracy of the online dataset is improved by 21% and 41% in the two directions.
This paper is interesting and it is quite well structured. I can recommend this for publication once the following suggestions are applied:
· The authors argue in the introduction: “Global Navigation Satellite System (GNSS) and Inertial Navigation System (INS) have highly complementary characteristics, which can effectively overcome the adverse effects of the harsh environment with the two systems integrated [1-3]. GNSS/INS integrated navigation has been widely used in high-precision positioning fields such as vehicle navigation, mobile mapping systems, and aerial photogrammetry [4].” Most of these potential applications are imprecise and refer to very broad fields of knowledge. In addition, citation 4 refers to the study of Grejner-Brzezinska, D. A.; Da, R.; Toth, C. (1998). However, to my understanding, this is quite an old study and so many advances in the combined used and potential applications of GNSS and GIS systems were developed. After that, the authors argue “Moreover, because the GNSS/INS tightly coupled system can maintain continuous navigation and positioning despite the lack of visible satellites, it has been extensively recognized by many scholars [5-7].” I recommend detail and referring to more particular and significant studies and applications. Just, for example, Balsa-Barreiro (2015) made a very extensive literature review of the combination of GPS and INS(IMU) systems for dynamic vehicle positioning (“Application of GNSS and GIS to road infrastructures: a study on naturalistic driving”).
· Just a few lines ahead, the authors argue: “In the application of vehicle dynamic positioning, continuous, reliable, and high-precision GNSS/INS integrated observation information is urgently required.” I recommend including some studies where it is commented on this issue. Just, for example, the formerly mentioned author analyzed the effect of positioning problems in naturalistic driving experiments. He developed a research line and implemented a method for overcoming the problems related to the lack of positioning data in studies such as “Geo-referencing naturalistic driving data using a novel method based on vehicle speed”.
· Again, the authors argue “However, the integrated positioning performance is inevitably plagued by various outliers in the complex urban scenario [8].” Citation 8 refers to “Alqurashi, M.; Wang, J. Performance analysis of fault detection and identification for multiple faults in GNSS and GNSS/INS 566 integration. Journal of Applied Geodesy, 2015, 9(1), 35-48.”. My question is what is the relationship between the aim of this paper and the complexity of urban scenarios? Do you mean that urban scenarios are complex because of their morphology/structure that limits the adequate reception of at least four satellites all the time?
· “For the GNSS receiver, the multipath effect and NLOS signal are the commonly abominable factors to restrict the positioning performance of GNSS, which further subsides the GNSS/INS integration system positioning capability [9, 10].” -> I think “abominable” is not the most scientific word to use in this context.
· “Due to the relatively low cost and the ability to provide accurate time reference and absolute 57 coordinate information, the GNSS sensor is still an indispensable core component of most navigation and positioning system.” -> You mean “systems” at the end.
· “We need to mitigate the error of the standalone GNSS positioning with innovative signal processing methods to promote the performance of GNSS/INS integration.” Why the authors does not consider the combined use of “odometers” instead of INS systems?
· “However, the performance of GNSS/INS integration suffers from excessive unexpected GNSS outliers such as multipath/NLOS signal in dense urban areas”. After that, the authors argue than their method “does not rely on external geographic information data and other sensors, which has better practicability and environmental adaptability.” My question is if the combined use of Building Information Models (BIMs) or Digital Surface Models (DSMs) in highly urbanized areas (and urban canyons) could optimize by increasing accuracies and liableness to your method?
· About the graphical part, I am missing an adequate cartographic treatment of the screenshots shown in Figures 4 and 10, by including a graphical scale. In Figure 7, I am missing the units used in the axis, and in Figure 13 one of the labels is in the reverse position.
